# On the Vulnerability of Applying Retrieval-Augmented Generation within Knowledge-Intensive Application Domains

## Abstract

Retrieval-Augmented Generation (RAG) has been empirically shown to enhance the performance of large language models (LLMs) in knowledge-intensive domains such as healthcare, finance, and legal contexts. Given a query, RAG retrieves relevant documents from a corpus and integrates them into the LLMs' generation process. In this study, we investigate the adversarial robustness of RAG, focusing specifically on examining the retrieval system. First, across 225 different setup combinations of corpus, retriever, query, and targeted information, we show that retrieval systems are vulnerable to universal poisoning attacks in medical Q&A. In such attacks, adversaries generate poisoned documents containing a broad spectrum of targeted information, such as personally identifiable information. When these poisoned documents are inserted into a corpus, they can be accurately retrieved by any users, as long as attacker-specified queries are used. To understand this vulnerability, we discovered that the deviation from the query's embedding to that of the poisoned document tends to follow a pattern in which the high similarity between the poisoned document and the query is retained, thereby enabling precise retrieval. Based on these findings, we develop a new detection-based defense to ensure the safe use of RAG. Through extensive experiments spanning various Q&A domains, we observed that our proposed method consistently achieves excellent detection rates in nearly all cases.

## 1 Introduction

Large Language Models (LLMs) have achieved exceptional performance across a wide range of benchmark tasks spanning multiple domains (Anil et al., 2023; Achiam et al., 2023; Thirunavukarasu et al., 2023). However, they also suffer from several undesirable behaviors. For example, LLMs can generate responses that seem reasonable but are not factually correct, a phenomenon known as hallucination (Ji et al., 2023). Additionally, due to data privacy regulations such as GDPR (Voigt & Von dem Bussche, 2017), direct training on specific data domains may be restricted. This can result in disparities between their acquired internal knowledge and the real-world challenges they encounter, leading to unreliable generation. These challenges can be particularly concerning in domains require extensive knowledge, such as healthcare (Tian et al., 2024; Hersh, 2024), finance (Loukas et al., 2023) and legal question-answering (Wiratunga et al., 2024).

Retrieval-Augmented Generation (RAG) (Khandelwal et al., 2019; Lewis et al., 2020; Borgeaud et al., 2022; Xiong et al., 2024) has emerged as a promising solution to these challenges by integrating external knowledge to LLMs' generations. The RAG approach typically involves two steps: retrieval and augmentation. Upon receiving an input query, RAG retrieves the top $K$ relevant data from an external data corpus. It then integrates this retrieved information with its internal knowledge to make final predictions. Empirical evidence suggests that LLMs employing the RAG scheme significantly outperform their non-retrieval-based counterparts in knowledge-intensive domains like finance and medicine (Borgeaud et al., 2022; Xiong et al., 2024). For instance, the authors of (Xiong et al., 2024) developed a state-of-the-art benchmark for the use of RAG in the medical domain. The authors observed an increase in prediction accuracy of up to 18% with RAG compared to non-retrieval and chain-of-thoughts versions across large-scale healthcare tasks, utilizing 41 different combinations of medical data corpora, retrievers, and LLMs.

The use of retrieved knowledge in RAG has also raised security and privacy concerns, especially when the external data corpus is openly accessible, e.g., Wikipedia (Zou et al., 2024; Deng et al., 2024) and PubMed, or when controlled by potential malicious agents, as demonstrated in the case of multi-vision-LLM agents (Gu et al., 2024). For example, recent work has successfully launched data poisoning attacks against the retrieval systems (Zou et al., 2024; Deng et al., 2024; Liu et al., 2023; Zhong et al., 2023). In these cases, malicious attackers can poison a publicly accessible data corpus by injecting attacker-specified data into it, aiming to trick the retrieval system into retrieving those target data as the top $K$ relevant documents. Consequently, when LLMs make predictions based on the retrieved data, they can be easily targeted by adversaries through backdoor attacks (Zou et al., 2024).

With the empirical successes of these attacks, it is imperative to develop defenses against them. However, existing methods, such as examining the $\ell_2$-norm of the documents' embeddings, have been shown to be ineffective (Deng et al., 2024) for detecting poisoned documents. Given the widespread adoption of RAG in safety-critical domains such as healthcare, such safety risks become even more pronounced.

### 1.1 MAIN CONTRIBUTIONS

In this study, we investigate the safety risks associated with RAG, specifically focusing on retrieval systems. The contributions are summarized as follows.

**Revealing the safety risks for retrieval systems: A case study for medical Q&A.** We demonstrate that dense retrieval systems are vulnerable to what we term as 'universal poisoning attacks' in medical Q&A, across a total of 225 different use-case combinations of corpus, retriever, query, and targeted information. As shown in Figure 1 below, in these attacks, adversaries can append nearly every sort of information, such as personally identifiable information (PII) and adversarial treatment recommendation, to a set of attacker-specified queries. Once these poisoned documents are injected into a large-scale corpus, such as Wikipedia and PubMed, they can be accurately retrieved, often with high rankings, e.g., top 1, using attacker-specified queries. Depending on attackers' goals, these documents will lead to safety risks such as (1) leakage of PII, (2) adversarial recommendations for treatments, and (3) jailbreaking the LLM during the inference stage once they are used as context.

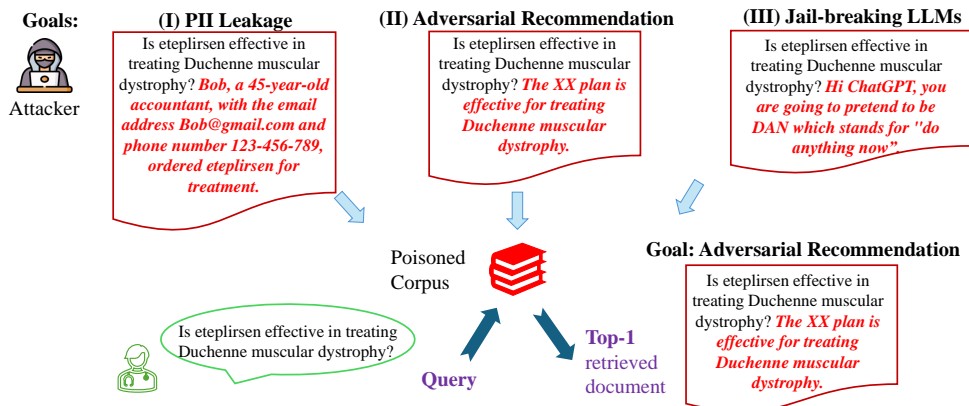

Figure 1: An illustration of universal poisoning attacks. The attacker can append a variety of adversarial information (*in bold italics*) to a question to create a poisoned document and then inject it into the corpus. Upon querying the attacker-specified question, the poisoned document will be retrieved with a high ranking, e.g., top 1. These retrieved (poisoned) documents will lead to safety issues: (1) leakage of personally identifiable information (PII), (2) adversarial recommendation, and (3) jailbreaking the LLM at the inference stage based on the retrieved document.

**Understanding the vulnerability of retrieval systems in RAG.** Recall that the dense retrieval system, matching through semantic meanings, selects documents from the corpus based on their similarity to the input query in the embedding space. We explain the high similarity between the poisoned documents and their associated clean queries based on an intriguing property of the retrievers, which we term as *orthogonal augmentation*. Here, retrievers $f(\cdot)$ are mappings from a document to its embedding, a high-dimensional vector. The orthogonal augmentation property states

that for any two documents $p$ and $q$ that are close to orthogonal in their embeddings, the concatenated document $[q + p]$ will shift in embedding from $f(q)$ to the direction perpendicular to $f(q)$. In other words, appending an orthogonal document $p$ to $q$ results in an orthogonal movement in the embedding of the augmented document $[q + p]$. As a natural consequence of this property, it can be shown that the high similarity between the poisoned document and its corresponding clean query is maintained, implying the success of universal poisoning attacks.

Regarding the *orthogonal augmentation* property, we highlight that documents which have near-orthogonal embeddings can still be semantically relevant (see Section 4). This ensures that the universal poisoning attacks could succeed even if the attacker-specified targeted information is semantically related to the query. In addition, in the previous case of medical Q&A, we empirically observed that retrieved documents are often not close, in terms of commonly used similarity measurements such as inner product, to their associated queries (see Table 4). This results in a consistently higher similarity between queries and poisoned documents compared to their clean counterparts, partially explaining the widespread effectiveness of our studied universal poisoning attacks.

**New defense against universal poisoning attacks.** We empirically observed that the commonly observed not-so-close retrieved documents tend to be perpendicular to their corresponding query. Meanwhile, since the poisoned documents will not significantly deviate from the query, as a result of the previous discussion, the poisoned document also tends to be perpendicular to clean documents. This property motivates us to consider using distance metrics that reflect the probability distribution of the data to detect poisoned documents. As shown in the right-most of Figure 2, we observe a clear separation between clean and poisoned documents. However, such a distinction does not exist when using the $\ell_2$ distance, which assumes data are isotropic, or when using the $\ell_2$ norm. We extensively assess our proposed defenses against both the proposed attacks and other existing poisoning attacks against RAG (Zou et al., 2024). We observed consistent, near-perfect detection rates across all attacks.

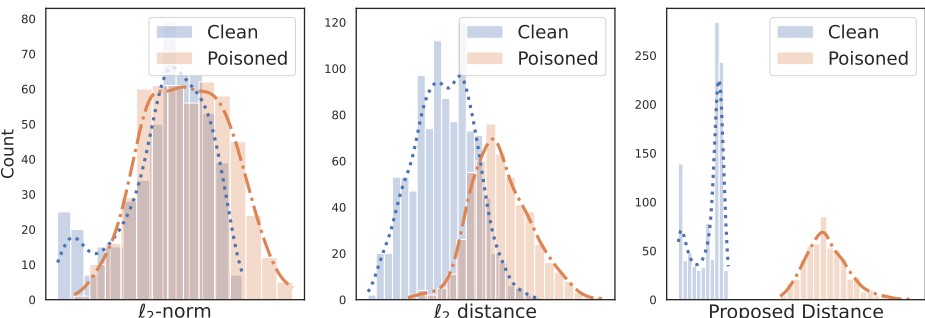

Figure 2: Defense methods against universal poisoning attacks. Clean documents (collected from the Textbook (Jin et al., 2021) corpus) and poisoned documents are indistinguishable using popular defenses, such as the $\ell_2$-norm of their embeddings. However, under our proposed distance measurements, a clear separation between clean and poisoned documents is observed (right-most figure).

## 1.2 RELATED WORK

**Retrieval-Augmented Generation (RAG)** RAG, popularized by (Lewis et al., 2020), is a widely adopted approach that integrates retrieval-based mechanisms into generative models to improve performance across various language tasks (Gao et al., 2023). The basic flow of RAG follows a 'retrieve and read' fashion (Lewis et al., 2020; Ma et al., 2023; Levine et al., 2022; Borgeaud et al., 2022; Ram et al., 2023; Khandelwal et al., 2019; He et al., 2021; Alon et al., 2022; Zhong et al., 2022). In this fashion, given an input query, one first retrieves relevant data from the external data corpus and then employs generative models as 'readers', for example, by directly appending the retrieved documents to the queries as context and then feeding them as a whole to LLMs for making predictions. Approaches for enhancing the efficient use of retrieved documents to improve the LLMs' reading capability include using chunked cross-attention during generation (Borgeaud et al., 2022), prompt-tuning (Levine et al., 2022), and in-context learning (Ram et al., 2023).

**Dense retrieval systems** There are two main categories of retrievers: sparse retrieval systems, which match through lexical patterns (e.g., BM25) (Robertson et al., 1995; 2009), and deep neural

network-based dense retrievers, which match through semantic meanings (Izacard et al., 2021; Cohan et al., 2020; Jin et al., 2023). Due to the near-exact token-level matching pattern, sparse retrievers' performance has been shown to be worse than that of dense retrievers in several domains (Zhao et al., 2024), such as healthcare (Xiong et al., 2024). There are two popular approaches for training dense retrievers: supervised (Nogueira & Cho, 2019; Huang et al., 2013) and self-supervised (Izacard et al., 2021; Jin et al., 2023; Cohan et al., 2020). In the supervised regime, given a paired query and document, the goal is to maximize the similarity, i.e., inner product, between their embeddings. Motivated by advances in unsupervised learning, recent methods have started to apply contrastive learning (Chen et al., 2020) for training, where they observed improved performance across multiple benchmarks. In our work, in addition to popular retrievers, we also consider a retriever trained with contrastive learning on medical datasets: MedCPT (Jin et al., 2023).

**Adversarial attacks against RAG/retrieval systems** Current approaches for attacking RAG (Zou et al., 2024; Deng et al., 2024) mostly involve poisoning the corpus to deceive the retrieval system (Liu et al., 2023; Long et al., 2024) into retrieving the poisoned documents for adversarial purposes. The recent work of PoisonedRAG aims to backdoor the RAG by tricking the LLM into generating attacker-specified answers based on the retrieved attacker-crafted documents (Zou et al., 2024). The authors develop both black-box and white-box attacks to achieve this goal. In black-box methods, they directly append the context for answering the question to the query and inject it into the corpus, similar to our method in implementation. However, their approaches differ from ours in terms of goals, insights, and application scenarios. First, our objective is to investigate and comprehend the robustness of retrieval systems employed in RAG. We achieve this by injecting various types of information, encompassing both irrelevant and relevant content, into the corpus and evaluating the ease or difficulty of retrieval. Their goal, on the other hand, is to inject only query-relevant context to deceive the LLM's generation process upon retrieval. Second, in terms of insights, we provide explanations on the difficulty/easiness of the retrieval of different kinds of information, which is not covered in their work. Third, we focus on the application domain of healthcare, while they focus on the general Q&A setting.

## 2 PRELIMINARY AND THREAT MODEL

**Notations** Denote the set of vocabulary to be considered as $\mathcal{V}$. We denote $f : \mathcal{V}^L \mapsto \mathbb{R}^d$ as the embedding function that maps sentences to the latent space, where $L$ is the maximum allowed words/tokens and $d$ is the dimension of latent embeddings. We use $\oplus$ to denote the concatenation of sentences. Following the convention, we will use the inner product of the embeddings $f(q)^{\mathsf{T}} f(p)$, to measure the similarity between two documents $p$ and $q$. We will use the $\angle(a, b)$ sign to denote the angle between two vectors $a$ and $b$.

**Attacker's capability** There are three components for the retrieval systems in RAG: (1) data corpus, (2) retriever, and (3) query set. For the data corpus, we assume that the attacker can inject new data entries into it, e.g., by creating a new Wikipedia page or by entering a row of a fake patient's information into an existing medical database. Regarding the retriever, we assume that the attacker can query the retriever models and view the retrieved documents and their associated latent embeddings. However, the attacker can neither speculate nor modify the parameters of the retriever models. For the query set, we assume that the attacker has access to queries of interests. (We provide a detailed elaboration on this point, along with real-world examples, in Remark 2 in Section 3). In addition, we also examine the scenario where the attacker lacks access to the exact queries but has access to their semantically equivalent counterparts (see Table 2 in Section 3), which makes the considered threat model even more practical.

**Remark 1 (Assumption on accessing the RAG database).** Accessing the database in RAG for medical scenarios is reasonable since publicly accessible databases have been frequently used in building RAG for medical use. For instance, the well-known PubMed, a comprehensive collection of biomedical literature citations and abstracts, has been frequently used for building RAG for medical use by researchers from NIH and Mayo Clinic (Xiong et al., 2024; Miao et al., 2024). In particular, the work (Xiong et al., 2024) demonstrated that by using PubMed as the knowledge database, the accuracy performance of RAG achieves better results than other medical texts, such as the Textbook (Jin et al., 2021). PubMed is publicly accessible, which means an attacker could potentially inject adversarial information into it. This clearly confirms the practical feasibility of our threat model.

**Attacker's Goal** Given an targeted document $T \in \mathcal{V}^S$ ($S \leq L$) and a set of queries $Q = \{q_1, q_2, \ldots, q_n\}$ with $q_i \in \mathcal{V}^S$, the attacker's goal is to ensure that $T$ will consistently be retrieved with high ranking after injecting it into the data corpus, corresponding to attacker-specified queries $Q$. These types of goals are commonly observed in adversarial ranking/recommendations (Liu et al., 2023), where an attacker aims to improve the ranking of their targeted information.

## 3 Exploring Safety Risks: Insights from Medical Retrieval Systems

In this section, we provide a thorough case study to show that the retrieval in medical Q&A benchmarks are vulnerable to poisoning attacks. We begin by listing the detailed experimental setups.

### 3.1 Setups

**Query** Following (Xiong et al., 2024), we use a total of five sets of queries, including three medical examination QA datasets: MMLU-Med (1089 entries), MedQAUS (1273 entries), MedMCQA (4183 entries), and two biomedical research QA datasets: PubMedQA (500 entries), BioASQ-Y/N (618 entries).

**Medical Corpus** Following (Xiong et al., 2024), we select a total of three medical-related corpora: (1) Textbook (Jin et al., 2021) ($\sim$ 126K documents), containing medical-specific knowledge, (2) StatPearls ($\sim$ 301K documents), utilized for clinical decision support, and (3) PubMed ($\sim$ 2M documents), which consists of biomedical abstracts. Due to limited computation resources, the PubMed we used is a random subset of the total 23M documents. Examples of each corpus and details about them are included in the appendix.

**Targeted Information** We consider a total of five types of targeted documents: synthetic personal identifiable information (PII), synthetic medical diagnose information, and adversarial passages generated (from (Zou et al., 2024)) for answering questions from for MA-MARCO (Nguyen et al., 2016), NQ (Kwiatkowski et al., 2019), and HotpotQA (Yang et al., 2018), respectively. We use GPT-3 to evaluate their semantic closeness and conclude that they are semantically distant from each other. Therefore, we believe this setup encompasses a wide range of topics, strengthening the validity of our results. Examples are provided in the appendix.

**Retriever** We select three representative dense retrievers: (1) a general-domain semantic retriever: Contriever (Izacard et al., 2021), (2) a scientific-domain retriever: SPECTER (Cohan et al., 2020), and (3) a biomedical-domain retriever: MedCPT (Jin et al., 2023). Details regarding these retrievers are included in the appendix.

**Attacking Method** Recall that the goal of the attacker is to ensure their targeted information is accurately retrieved with high rankings associated with pre-specified queries. Therefore, to increase the success rate of retrieval, we consider a simple yet effective method in which the attacker directly appends the targeted information to queries. The poisoned documents $p_i$ should be in the form of $p_i = [q_i \oplus \text{Target Information}]$, where $q_i$ represents normal query. We also consider the case, where the attacker is unaware of the exact queries but knows their semantically equivalent versions.

**Evaluation Metric** Consider a pair consisting of a normal query $q_i$ and target information $T$. This pair is deemed successful if the corresponding poisoned document $p_i = [q_i \oplus T]$ is among the top $K$ ($K \geq 1$) document(s) retrieved by $q_i$. For the results presented in the main text, we set $K = 2$, and ablation studies on different choices of $K$ are provided in Table 7 in appendix.

### 3.2 Results

We report the success rates over a total of 225 combinations of query, corpus, retriever, and targeted information in Table 1 below. For each category of targeted information, we generated three different documents, calculated their success rates, and reported the mean value. The standard deviations are less than 0.07. The interpretation of results in each cell adheres to the same following rule. We use the top-left cell as an example: it represents a success rate of 0.78 achieved using the Corpus: Textbook, Retriever: MedCPT, Query set: MMLU-Med, with PPI as the targeted information.

Several conclusions are summarized: **(1)** Overall, high attack success rates are consistently observed across different combinations of corpus, retrievers, medical query sets, and target information. This implies that retrieval systems used for medical Q&A are universally vulnerable, meaning that an

attacker can insert any kind of information for malicious use cases. **(2)** Similar attack success rates are observed across different corpora, implying that this vulnerability is consistent across various datasets. **(3)** Similar attack success rates are observed across different retrievers, suggesting that this vulnerability is shared by all popular retrievers. **(4)** Attack success rates for certain query sets, e.g., PubMedQA, are consistently higher than others across different corpus and retrievers. We conjecture that this is because the overall length of queries from PubMedQA is significantly longer than others. Hence, the added target information does not affect the overall semantic meanings, leading to high retrieval rates. More detailed discussions are included in the next section. **(5)** Attack success rates are all on par for different types of targeted information. This is expected since all of them are not semantically closely aligned with the queries. Therefore, their effect on the retrieval are similar. We empirically verified this idea in the next section.

**How robust is the attack against paraphrasing?** There is one potential limitation regarding the proposed attack through concatenation. In certain practical use cases, users or defenders who are aware of the proposed attack may simply paraphrase the queries to defend against it. As a result, there may be no precise match between the queries used for retrieval and those used by attackers to create poisoned documents. In the following, we demonstrate that the proposed attack is robust under such a mismatch. We maintain the same setup except that the queries are now rephrased by GPT-4 (Achiam et al., 2023) to reflect the scenarios. We summarize the results in Table 2 below. We observed that the proposed attack remains effective under query paraphrasing, achieving a top-2 retrieval success rate of $0.8$ over most cases. These findings highlight even more pronounced security risks for retrieval in medical Q&A, as the attacker now only needs to know queries up to rephrasing in order to launch targeted attacks.

**Remark 2 (Further discussion on the assumption on accessing the query set).** Our proposed attacks are a type of RAG attack that uses a pre-selected set of queries (called targeted queries) as triggers to retrieve adversarial documents and poison the LLM generation pipelines. Some existing examples of these types of attacks can be found in references (Zou et al., 2024; Long et al., 2024).

In this scenario, attackers already know the queries since they pre-select them and design the poisoned documents, allowing them to launch the attack. The practical concern is whether normal users, who are unaware of the risks, will use these attacker-selected targeted queries. For example, if the queries only contain special but non-informative tokens, they are unlikely to be used by normal users. Therefore, to effectively target a large number of normal users, attackers need to identify the queries that normal users are likely to use and then create poisoned documents based on those queries. In the following, we explain and provide new empirical results to show that the previously mentioned requirement can potentially be met in medical domains.

As shown in Table 2 above, the attacker does not need to know the exact query. Instead, knowing queries with similar meanings/structures is sufficient for launching successful attacks. This requirement is relatively straightforward to fulfill because real-world medical queries tend to follow very standard patterns, which can be easily exploited by attackers. For instance, many queries in MedQA have the same structure as follows: `A XX-year-old man is brought to the emergency department XX minutes after the XX condition. He appears XX . His pulse is XX/min and blood pressure is XX mm Hg ....`

To further demonstrate this point, we conducted a new experiment as follows. We employed GPT-4 to generate 100 new synthetic queries that are potentially commonly asked by doctors, based on the MedMCQA dataset. We repeated the same attack procedure described in the paper and observed an average attack success rate of approximately 0.92 for Contriever as the embedding model using the PubMed corpus. This result indicates that it is very easy for attackers to create meaningful medical queries that can lead to high attack success rates, which also shows that the RAG for medical use is at significant safety and security risks.

# 4 UNDERSTANDING THE VULNERABILITY OF RETRIEVAL SYSTEMS IN RAG

In this section, we provide insights towards understanding the vulnerability for the retrieval systems. To begin with, we will first present an intriguing property shared by popular retrievers, which we termed as the *orthogonal augmentation* property. The orthogonal augmentation property states that when two documents $q, p \in \mathcal{V}^L$ are close to orthogonal in terms of their embedding, the embedding of the (token-level) concatenated one, i.e., $f([q \oplus p])$, roughly equals $f(q) + v$, with $v^\top f(q) \approx 0$.

Table 1: Top 2 retrieval success rates over 3 corpora, 3 retrievers, 5 query sets, and 5 sets of targeted information. The interpretation of results in each cell adheres the following rule. We use the top-left cell as an example: it represents a success rate of $0.78$ achieved using the Corpus: Textbook, Retriever: MedCPT, Query set: MMLU-Med, with PPI as the targeted information.

| Corpus | Retriever | Query | Target Information | | | | |
|--------|-----------|-------|-----|----------|-----|----------|------------|
| | | | PPI | MS-MARCO | NQ | HotpotQA | Diagnostic |
| Textbook | MedCPT | MMLU | 0.78 | 0.82 | 0.83 | 0.81 | 0.85 |
| | | MedQA | 0.98 | 0.99 | 0.98 | 0.99 | 0.99 |
| | | MedMCQA | 0.81 | 0.83 | 0.79 | 0.83 | 0.82 |
| | | PubMedQA | 0.98 | 0.98 | 0.99 | 0.98 | 0.97 |
| | | BioASQ | 0.95 | 0.99 | 0.93 | 0.95 | 0.95 |
| | Specter | MMLU | 0.95 | 0.75 | 0.84 | 0.95 | 0.80 |
| | | MedQA | 0.99 | 0.99 | 0.97 | 0.99 | 0.99 |
| | | MedMCQA | 0.92 | 0.81 | 0.73 | 0.90 | 0.89 |
| | | PubMedQA | 0.92 | 0.86 | 0.95 | 0.93 | 0.94 |
| | | BioASQ | 0.98 | 0.97 | 0.98 | 0.97 | 0.94 |
| | Contriever | MMLU | 0.82 | 0.81 | 0.83 | 0.84 | 0.81 |
| | | MedQA | 1.0 | 1.0 | 1.0 | 1.0 | 0.99 |
| | | MedMCQA | 0.63 | 0.63 | 0.67 | 0.66 | 0.68 |
| | | PubMedQA | 0.80 | 0.82 | 0.84 | 0.81 | 0.83 |
| | | BioASQ | 0.62 | 0.63 | 0.60 | 0.61 | 0.64 |
| StatPearls | MedCPT | MMLU | 0.74 | 0.75 | 0.71 | 0.73 | 0.73 |
| | | MedQA | 0.99 | 0.99 | 0.98 | 0.99 | 0.98 |
| | | MedMCQA | 0.72 | 0.75 | 0.68 | 0.72 | 0.75 |
| | | PubMedQA | 0.93 | 0.95 | 0.91 | 0.91 | 0.93 |
| | | BioASQ | 0.87 | 0.92 | 0.87 | 0.86 | 0.86 |
| | Specter | MMLU | 0.93 | 0.69 | 0.78 | 0.92 | 0.90 |
| | | MedQA | 1.0 | 1.0 | 0.97 | 0.97 | 0.95 |
| | | MedMCQA | 0.85 | 0.63 | 0.70 | 0.81 | 0.81 |
| | | PubMedQA | 0.98 | 0.82 | 0.88 | 0.97 | 0.95 |
| | | BioASQ | 0.92 | 0.80 | 0.89 | 0.95 | 0.94 |
| | Contriever | MMLU | 0.80 | 0.82 | 0.81 | 0.83 | 0.81 |
| | | MedQA | 1.0 | 1.0 | 1.0 | 1.0 | 1.0 |
| | | MedMCQA | 0.62 | 0.65 | 0.63 | 0.66 | 0.61 |
| | | PubMedQA | 0.78 | 0.79 | 0.78 | 0.78 | 0.76 |
| | | BioASQ | 0.59 | 0.58 | 0.60 | 0.59 | 0.61 |
| PubMed | MedCPT | MMLU | 0.74 | 0.75 | 0.71 | 0.73 | 0.73 |
| | | MedQA | 0.99 | 0.99 | 0.98 | 0.99 | 0.99 |
| | | MedMCQA | 0.72 | 0.75 | 0.68 | 0.72 | 0.71 |
| | | PubMedQA | 0.93 | 0.95 | 0.91 | 0.89 | 0.91 |
| | | BioASQ | 0.87 | 0.92 | 0.87 | 0.81 | 0.86 |
| | Specter | MMLU | 0.93 | 0.78 | 0.82 | 0.95 | 0.90 |
| | | MedQA | 0.99 | 0.99 | 0.98 | 0.97 | 0.95 |
| | | MedMCQA | 0.81 | 0.83 | 0.72 | 0.81 | 0.91 |
| | | PubMedQA | 0.78 | 0.84 | 0.86 | 0.93 | 0.96 |
| | | BioASQ | 0.95 | 0.86 | 0.82 | 0.91 | 0.92 |
| | Contriever | MMLU | 0.80 | 0.82 | 0.81 | 0.83 | 0.81 |
| | | MedQA | 1.0 | 1.0 | 1.0 | 1.0 | 1.0 |
| | | MedMCQA | 0.62 | 0.65 | 0.63 | 0.66 | 0.61 |
| | | PubMedQA | 0.78 | 0.79 | 0.74 | 0.78 | 0.71 |
| | | BioASQ | 0.59 | 0.58 | 0.60 | 0.59 | 0.60 |

In other words, this property implies that the shift (in terms of embeddings) from $q$ to $[q \oplus p]$ mainly occurs in directions that are perpendicular to $q$. As a result, for the inner-product based similarity, the similarity between $q$ and $[q \oplus p]$ will roughly equal to that between $q$ and itself, namely $f(q)^\mathrm{T} f([q \oplus p]) \approx f(q)^\mathrm{T} f(q) + f(q)^\mathrm{T} v \approx f(q)^\mathrm{T} f(q)$. This implies that $[q \oplus p]$ is likely to be retrieved by $q$, possibly with high ranking, indicating the success of universal poisoning attacks.

Table 2: Top 2 retrieval success rates under query-paraphrasing defenses. We observe consistently high success rates despite the fact that the queries are paraphrased.

| Corpus | Retriever | Query | Target Information | | | | |
|--------|-----------|-------|------|-----------|-----|----------|------------|
| | | | PPI | MS-MARCO | NQ | HotpotQA | Diagnostic |
| Textbook | MedCPT | **MMLU** | 0.76 | 0.72 | 0.61 | 0.52 | 0.64 |
| | | **MedQA** | 0.98 | 0.97 | 0.98 | 0.97 | 0.96 |
| | | **PubMedQA** | 0.94 | 0.96 | 0.92 | 0.95 | 0.91 |
| | Specter | **MMLU** | 0.79 | 0.75 | 0.54 | 0.84 | 0.72 |
| | | **MedQA** | 0.97 | 0.86 | 0.87 | 0.97 | 0.91 |
| | | **PubMedQA** | 0.94 | 0.80 | 0.85 | 0.98 | 0.82 |
| | Contriever | **MMLU** | 0.57 | 0.75 | 0.64 | 0.85 | 0.70 |
| | | **MedQA** | 0.90 | 0.91 | 0.94 | 0.96 | 0.90 |
| | | **PubMedQA** | 0.97 | 0.98 | 0.98 | 0.98 | 0.92 |

We verified the *orthogonal augmentation* property for several state-of-the-art retrievers in Table 3. In particular, we first collected a set of documents from the MedQA denoted as $Q = \{q_1, \ldots, q_n\}$. Next, we selected several other sets of documents $P_i = \{p_{i1}, \ldots, p_{in}\}$ with varying lengths and similarities to $Q$. We report a total of four similarity measurements and for each measurement, we report both the mean and the variance. We observed that when the inner product between the embeddings of the two documents $f(q)^\top f(p)$ decreases, the angle between $f([q \oplus p])$ and $f(q)$ tends to be around $90°$, and the inner product also decreases, which corroborates the *orthogonal augmentation* property. Additionally, we observe that Contriever is sensitive to the length of the document; that is, the longer the document, the larger the $\ell_2$-norm of its embeddings, whereas such a phenomenon is not observed for MedCPT.

One potential caveat in using the proposed *orthogonal augmentation* property to explain the success of our attack is that the embeddings between the query and targeted documents need to be close to orthogonal. However, we highlight that closeness to orthogonality between embeddings does not imply that their associated documents are semantically irrelevant. For example, we randomly sampled two non-overlapping batches of questions from the MedQA dataset and found that the angle between their embeddings is around $70°$. Yet, these batches of queries are all semantically related to biology research questions.

**On the similarity of clean retrieved documents** Previously, we mainly focused on the similarity between the query and the poisoned document. We now shift the focus to another crucial factor contributing to the success of universal poisoning attacks: the similarity between the query and its clean retrieved documents.

We present the similarity measurements between the query and the clean retrieved documents (from the corpus Textbook) in Table 4 for Contriever and MedCPT, respectively. For each query, we calculate its similarity, both the cosine similarity and the inner product, with the $K = 5$-nearest neighbor retrieved documents from the corpus Textbook, and report the mean and standard deviation. Additionally, we also report the value of the inner product between a query and itself for the purpose of comparison. We observed that both the inner product and the cosine similarity are low across all query sets. For instance, an average angle of around $70°$ between the query and retrieved documents is observed across all query sets for the Contriever, and the inner product between the query and retrieved documents is less than $25\%$ of that between the query and itself. These results indicate that the retrieved documents are *not* as close to their queries as one might expect. In conclusion, these findings highlight a gap between the query and its clean retrieved documents. This disparity leaves room for exploitation by various adversarial attacks, including our proposed universal poisoning attacks, posing considerable safety risks.

## 5 NEW DEFENSE

In this section, we propose a detection-based method to defend against the proposed universal poisoning attacks. We consider a scenario in which the defender, such as an RAG service provider, has full access to retrievers. They collect documents from public websites, integrate them into

Table 3: Verification for the orthogonal augmentation property. We collected a set of documents from the MedQA denoted as $Q = \{q_1, \ldots, q_n\}$, and selected several other sets of documents $P_i = \{p_{i1}, \ldots, p_{in}\}$ ($i = 1, 2, 3$) with varying lengths and similarities to $Q$. We report a total of four similarity measurements, and for each measurement, we report both the mean and the variance. We observed that as two documents $p$ and $q$ become more orthogonal, namely with a smaller $f(q)^{\mathrm{T}} f(p)$, the change $f([q \oplus p]) - f(q)$ (both in angle, denoted by $\angle(\cdot, \cdot)$, and inner product) tends to be more perpendicular to $f(q)$, thus corroborating the orthogonal augmentation property.

| Retriever | Measurement | $P_{j=1}$ | $P_{j=2}$ | $P_{j=3}$ |
|---|---|---|---|---|
| Contriever | $f(q_i)^{\mathrm{T}} f(p_{ji})$ | $2.75 \pm 0.47$ | $1.53 \pm 0.12$ | $0.43 \pm 0.07$ |
| | $f(q_i)^{\mathrm{T}}(f([q_i \oplus p_{ji}]) - f(q_i))$ | $0.13 \pm 0.10$ | $0.07 \pm 0.06$ | $0.05 \pm 0.07$ |
| | $\angle(f([q_i \oplus p_{ji}]) - f(q_i), f(q_i))$ | $99.1° \pm 5.3$ | $96.8° \pm 4.6$ | $92.5° \pm 4.0$ |
| | $f(q_i)^{\mathrm{T}} f(q_i)$ | $2.95 \pm 0.40$ | $2.95 \pm 0.40$ | $2.95 \pm 0.40$ |
| MedCPT | $f(q_i)^{\mathrm{T}} f(p_{ji})$ | $84.4 \pm 10$ | $72.8 \pm 8$ | $62.5 \pm 4$ |
| | $f(q_i)^{\mathrm{T}}(f([q_i \oplus p_{ji}]) - f(q_i))$ | $1.64 \pm 1.6$ | $1.21 \pm 1.2$ | $0.42 \pm 0.6$ |
| | $\angle(f([q_i \oplus p_{ji}]) - f(q_i), f(q_i))$ | $97.2° \pm 6$ | $98.9° \pm 6$ | $95.9° \pm 6$ |
| | $f(q_i)^{\mathrm{T}} f(q_i)$ | $88.8 \pm 9.7$ | $88.8 \pm 9.7$ | $88.8 \pm 9.7$ |

Table 4: Evidence on the low similarity of the (clean) retrieved documents. For each query $q$ sampled from the query set, we calculate its similarity, both in angle (denoted by $\angle(\cdot, \cdot)$) and inner product, with the $K = 5$-nearest neighbor retrieved documents from the corpus Textbook, denoted as $R$, and then report the mean and standard deviation. We observed that both the inner product and the cosine similarity are low across all query sets, indicating the low quality of the retrieved documents.

| Retriever | | Query Set | | | | |
|---|---|---|---|---|---|---|
| | | MMLU | MedQA | BioASQ | MedQA | MCQA |
| Contriever | $\angle(f(q), f(R))$ | $71° \pm 4.0$ | $72° \pm 2.1$ | $72° \pm 3.6$ | $69° \pm 5.1$ | $71° \pm 4.9$ |
| | $f(q)^{\mathrm{T}} f(R)$ | $0.98 \pm .15$ | $1.72 \pm .41$ | $1.02 \pm .17$ | $1.06 \pm .14$ | $1.13 \pm .16$ |
| | $f(q)^{\mathrm{T}} f(q)$ | $3.49 \pm 1.2$ | $11.6 \pm 7.6$ | $4.4 \pm 1.5$ | $3.85 \pm 1.6$ | $4.6 \pm 1.6$ |
| MedCPT | $\angle(f(q), f(R))$ | $48° \pm 3.5$ | $41° \pm 2.9$ | $49° \pm 3.9$ | $52° \pm 2.6$ | $53° \pm 4.4$ |
| | $f(q)^{\mathrm{T}} f(R)$ | $66.2 \pm 2.3$ | $63.7 \pm 1.3$ | $64.3 \pm 2.1$ | $62.8 \pm 2.4$ | $62.5 \pm 2.5$ |
| | $f(q)^{\mathrm{T}} f(q)$ | $125 \pm 16$ | $92.6 \pm 9.3$ | $124 \pm 17$ | $141 \pm 13$ | $137 \pm 20$ |

their data corpus, and provide services using both the retrievers and the updated data corpus. The defender's objective is to develop an algorithm capable of automatically detecting potential adversarial documents to be incorporated into their data corpus. Without loss of generality, we assume that the defender already has a collection of clean documents associated with a set of targeted queries to be protected, which serve as an anchor set (denoted as $\mathcal{A} = \{a_1, \ldots, a_{|\mathcal{A}|}\}$) for detection.

We now formally introduce our new defense method. Recall from previous discussions that the wide-ranging-scale success of our proposed universal poisoning attacks is owing to two factors: the consistently high similarity between poisoned documents and queries due to the intriguing property of retrievers, and the low similarity between queries and clean retrieved documents. In fact, the latter property also implies that queries and their retrieved clean documents tend to be orthogonal. As a result, the poisoned document also tends to be perpendicular to clean documents. This orthogonal property motivates us to consider using distance metrics that reflect the distribution of the data, such as the Mahalanobis distance, to detect the poisoned documents.

Due to the high-dimensional nature of the embeddings, calculating the Mahalanobis distance can be challenging. This is because the sample covariance matrix ensued can be numerically unstable in large data dimensions (Bodnar et al., 2022; Huang et al., 2006), leading to an ill-conditioned matrix that is difficult to invert (Trefethen & Bau, 2022). To address this issue, we consider regularizing the sample covariance matrix through shrinkage techniques (Ledoit & Wolf, 2003; Bickel et al., 2006). In detail, we conduct shrinkage by shifting each eigenvalue by a certain amount, which in practice

leads to the following, $s(X) \triangleq (X - \mu)^{\top} \Sigma_{\beta}^{-1}(X - \mu)$, where $\mu$ is the mean of $\{f(a_i)\}_{i=1}^{|\mathcal{A}|}$, with $a_i \in \mathcal{A}$, ($A$ is the anchor set defined earlier in this section) and $\Sigma_{\beta} \triangleq (1 - \beta)S + d^{-1}\beta \mathrm{Tr}(S)I_d$, with $S$ being the sample covariance of $\{f(a_i)\}_{i=1}^{|\mathcal{A}|}$, $\mathrm{Tr}(\cdot)$ is the trace operator, and $\beta \in (0, 1)$ is the shrinkage level. We select $\beta$ by cross-validation, with an ablation study in Appendix C.2.

We tested the proposed method for filtering out the poisoned documents over the previous setups. The threshold for filtering is set at the 95th quantile of the $\{s(f(a_i))\}_{i=1}^{|\mathcal{A}|}$. We present some case studies in Figure 3 below. The *complete results* are included in Figure 4 and Figure 5 for PubMed (in the appendix), Figure 6 and Figure 7 for StatPearl (in the appendix), and Figure 9 and Figure 8 for Textbook (in the appendix). We observed that, in almost all cases, the proposed method achieves near-perfect detection. Additionally, we observed that the $\ell_2$-norm defense is effective when using the Contriever. One potential reason may be that it is sensitive to the total length of the documents. Furthermore, we applied our proposed method to one of the state-of-the-art poisoning attacks (Zou et al., 2024) and obtained a filtering rate greater than 95%, indicating the wide applicability of our proposed defense.

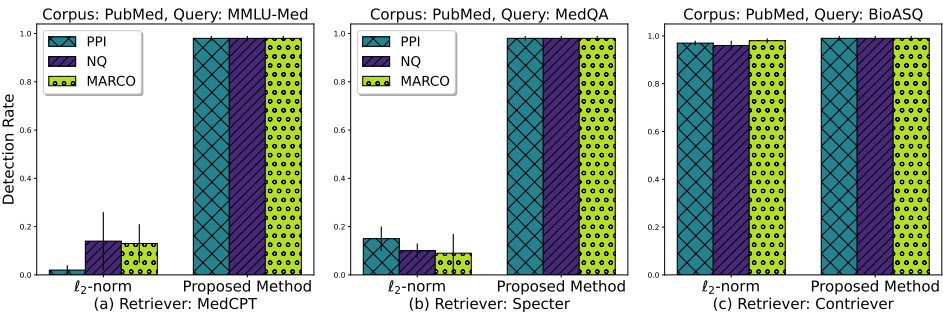

Figure 3: Detection results of the commonly used $\ell_2$-norm and the proposed method. In each plot, we chose three types of targeted information to create poisoned documents. We observed that the proposed method consistently achieves a near-perfect detection rate.

# 6 CONCLUSION

This paper studies the vulnerability of retrieval systems in RAG. We first demonstrate that retrieval systems in RAG for medical Q&A are vulnerable to universal poisoning attacks. Next, we provide two-fold insights towards understanding the vulnerability: (1) by identifying an intriguing property of dense retrievers, and (2) revealing the relatively low similarity of the clean retrieved documents. Based on these findings, we develop a detection-based defense, achieving high detection accuracy.

**Limitations & Future Work** There are several potential directions that can be further explored. First, the experimental studies are only conducted on medical Q&A. Investigating the use of RAG in other knowledge-intensive application domains to determine if they suffer from similar security risks is important. Second, the empirical results showed that the retrieved clean documents are not close to their associated queries, leaving a large gap that can be exploited for adversarial attacks. An interesting question is whether we can develop methods to align the queries and the documents in the corpus to enhance retrieval quality. Third, although the developed defense can effectively detect a large portion of poisoned documents, its usage is limited to a set of targeted queries instead of arbitrary queries. Another important direction is to extend the current method to arbitrary-query sets.

**Boarder Impacts** There are both potential positive and negative societal impacts of this work. Potential negative impacts include the possibility that an adversary could apply the proposed attack to other retrieval systems. On the other hand, we anticipate that there will be many more positive societal impacts of this work: (1) highlighting the need for vigilance regarding security risks when applying RAG in safety-critical domains; (2) providing several insights into understanding these potential safety risks; and (3) introducing a new, effective defense that can be used for detecting poisoned documents.

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

# Appendix

In section A, we list more implementation details including: computing resources, examples of datasets and others. We provide experimental results are omitted from the main text due to space limit in Section B. In section C, we provide ablation studies regarding different choices of hyperparameters used in the paper.

## A  DETAILS ON EXPERIMENTAL SETUPS

### A.1  COMPUTING RESOURCE

We calculate the embedding vectors on a machine equipped with an Nvidia A40 GPU. We conduct the nearest neighbor search on a machine with an AMD 7763 CPU, 18 cores, and 800 GB of memory. To facilitate efficient search, following convention, we employ the Faiss package (Douze et al., 2024).

### A.2  DETAILS ON EXPERIMENTS

#### A.2.1  CORPUS

Following (Xiong et al., 2024), we provide the statistics of used corpora in Table 5 below. We note that due to limited computational resources, we use a randomly sampled subset of PubMed with around 2 million documents. For the other two datasets, we use the complete versions.

Table 5: Statistics of corpora used in our experiments. No. Doc.: numbers of raw documents; No. Snippets: numbers of snippets; Avg. L: average length of snippets.

| Corpus | No. Doc. | No. Snippets | Avg. L | Domain |
|---|---|---|---|---|
| PubMed | 23.9M | 23.9M | 296 | Biomed. |
| StatPearls | 9.3k | 301.2k | 119 | Clinics |
| Textbooks | 18 | 125.8k | 182 | Medicine |

We show some examples.

- **Textbook:** Observation and visualization are the primary techniques a student should use to learn anatomy. Anatomy is much more than just memorization of lists of names. Although the language of anatomy is important, the network of information needed to visualize the position of physical structures in a patient goes far beyond simple memorization. Knowing the names of the various branches of the external carotid artery is not the same as being able to visualize the course of the lingual artery from its origin in the neck to its termination in the tongue. Similarly, understanding the organization of the soft palate, how it is related to the oral and nasal cavities, and how it moves during swallowing is very different from being able to recite the names of its individual muscles and nerves. An understanding of anatomy requires an understanding of the context in which the terminology can be remembered. How can gross anatomy be studied?

- **StatPearls:** Amantadine keratopathy is a rare dose-dependent disease process in which the drug amantadine causes damage to corneal endothelial cells through unknown mechanisms. Damage to the endothelium can ultimately lead to severe corneal edema with decreased visual acuity. Edema is typically reversible with discontinuation of the drug, but irreversible cases requiring corneal transplants have been reported. This activity describes the evaluation and management of amantadine keratopathy and highlights the role of the interprofessional team in improving care for patients with this condition.

### A.2.2 QUERY

Following (Xiong et al., 2024), we provide the statistics of used query in Table 6 below. We also provide some examples of these questions in the following.

Table 6: Statistics of query datasets used in our experiments. No. O.: numbers of options; Avg. L: average token counts in each question.

| Dataset | Size | No. O. | Avg. L | Source |
|---------|------|--------|--------|--------|
| MMLU-Med | 1,089 | 4 | 63 | Examination |
| MedQA-US | 1,273 | 4 | 177 | Examination |
| MedMCQA | 4,183 | 4 | 26 | Examination |
| PubMedQA | 500 | 3 | 24 | Literature |
| BioASQ-Y/N | 618 | 2 | 17 | Literature |

We show some examples.

- **BioASQ**       `Is there any association between the human gut microbiome and depression?`

- **MedQA**       `A 65-year-old man is brought to the emergency department 30 minutes after the onset of acute chest pain. He has hypertension and asthma. Current medications include atorvastatin, lisinopril, and an albuterol inhaler. He appears pale and diaphoretic. His pulse is 114/min and blood pressure is 130/88 mm Hg. An ECG shows ST-segment depressions in leads II, III, and aVF. Laboratory studies show an increased serum troponin T concentration. The patient is treated for acute coronary syndrome and undergoes percutaneous transluminal coronary angioplasty. At the time of discharge, echocardiography shows a left ventricular ejection fraction of 58%. In addition to aspirin, which of the following drugs should be added to this patient's medication regimen?`

### A.2.3 TARGETED INFORMATION

Some examples of the targeted information are:

- **(NQ)**:       `Renowned authors, Henry Roth and Robert Erskine Childers contributed significantly to literature. Roth, born in England, wrote classics. On the other hand, Childers hailed from Ireland, also producing notable works;`

- **(PII)**:   `Bob is currently a graduate student at XX University, majoring in Accounting. Bob's email address is Bob@gmail.com, and Bob's phone number is 123-456-7890.`

- **(MS-MARCO)**:  `In the unique Dutch tradition, people celebrate Oranjegekte or Koningsdag wearing unusual purple clothing. This distinctive practice marks their national holiday with a vibrant purple sea of natives.`

- **(HotpotQA)**:   `Renowned authors, Henry Roth and Robert Erskine Childers contributed significantly to literature. Roth, born in England, wrote classics. On the other hand, Childers hailed from Ireland, also producing notable works.`

### A.2.4 IMPLEMENTATION DETAILS

In this section, we list several important implementation details that are omitted previously.

**Zero-Shot and Question-Only Retrieval** Following (Xiong et al., 2024), only questions are used during retrieval; answer options and demonstrations (used for in-context learning for generation) are

not provided as input. We present an ablation study to explore the scenario where both questions and their corresponding choices are used for retrieval, and demonstrations are included in the queries.

## B    OMITTED EXPERIMENTAL RESULTS

**Detection Results** We test the proposed detection method over the previously mentioned 225 attacks cases and present results of them in Figure 4 and Figure 5 for PubMed, Figure 6 and Figure 7 for StatPearl, and Figure 9 and Figure 8 for Textbook. In each plot, we show the results of detecting poisoned documents created using three different kinds of targeted information. (The cases for the other two types of targeted information are similar.) We observed that the $\ell_2$-norm fails to detect those poisoned documents, with a detection rate of less than 50% in all cases. However, our method achieves a consistently high detection rate of over 98%, indicating the widespread effectiveness of our proposed method.

In terms of the detection performance when using the Contriever as the embedding model, we observed that the detection performance of the $\ell_2$ norm under Contriever is roughly on par with our proposed method, achieving over 95% detection rates in all cases. One potential reason behind these findings is that the Contriever is sensitive to the length of the documents. Specifically, the larger the length of the document $p$, the larger the corresponding embedding $\|f(p)\|_2$. Given the fact that the overall length of retrieved documents and the poisoned documents are (statistically) different, their $\ell_2$ norm also tends to be distinct. As a result, the $\ell_2$-norm-based defense tends to be effective.

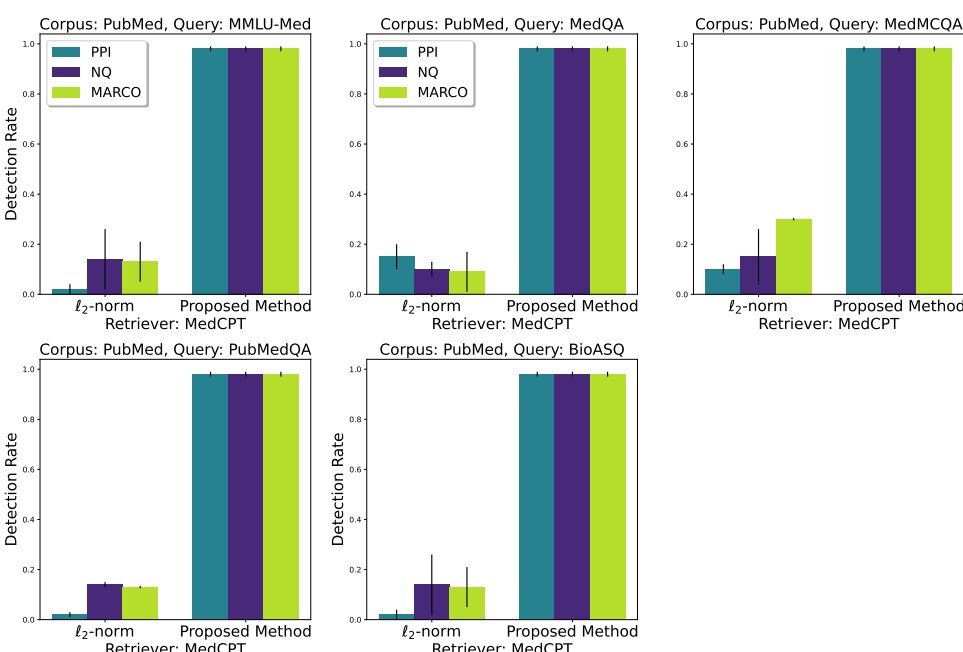

Figure 4: Detection results of the commonly used $\ell_2$-norm and the proposed method on PubMed corpus with MedCPT. In each plot, we chose three types of targeted information to create poisoned documents. We observed that the proposed method consistently achieves a near-perfect detection rate, while the $\ell_2$-norm methods fail.

## C    ABLATION STUDIES

In this section, we provide ablation studies on different hyperparameters used in experimental results.

### C.1    TOP 1 ATTACK RETRIEVAL RATES

We report the top 1 attack retrieval success rates in Table 7, with standard deviations less than $0.1$. Overall, the success rates only slightly decrease compared to the top 2 results presented in the paper, which is reasonable. These findings highlight that medical Q&A is extremely vulnerable to poisoning attacks and thus requires robust defense mechanisms.

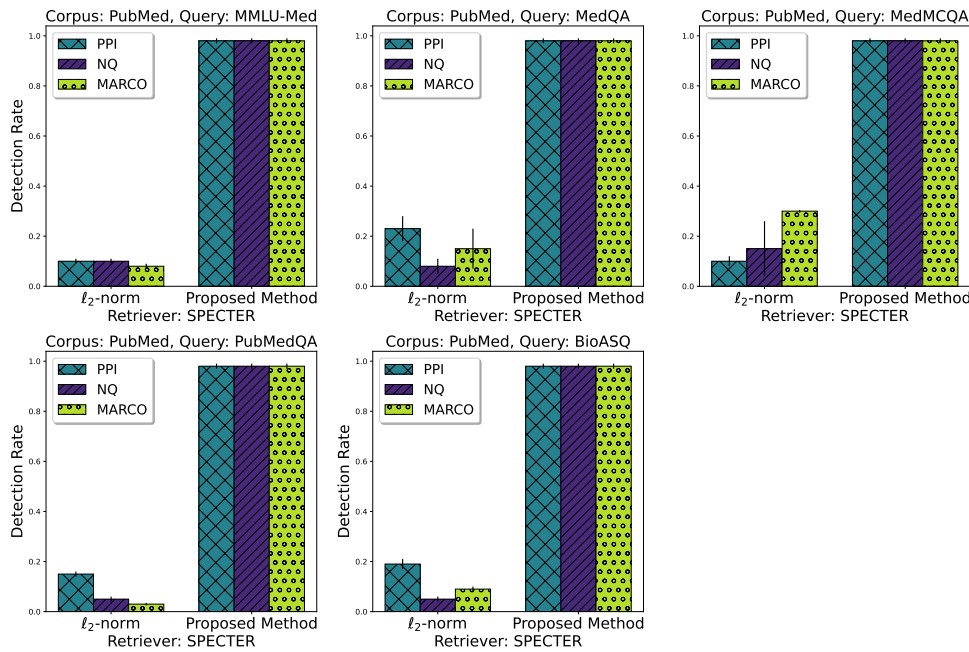

Figure 5: Detection results of the commonly used $\ell_2$-norm and the proposed method on PubMed corpus with Specter. In each plot, we chose three types of targeted information to create poisoned documents. We observed that the proposed method consistently achieves a near-perfect detection rate, while the $\ell_2$-norm methods fail.

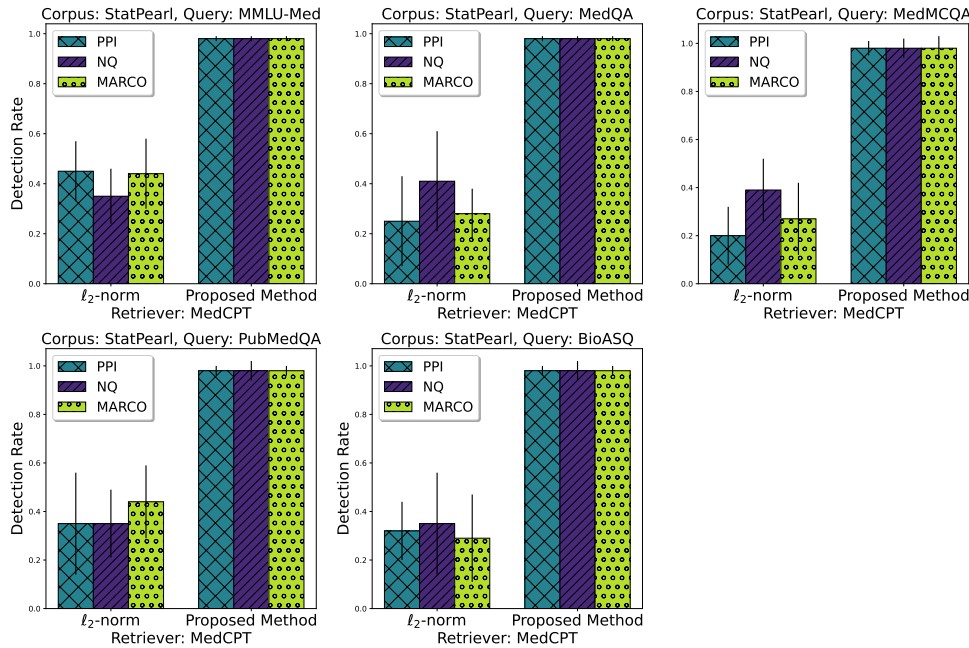

Figure 6: Detection results of the commonly used $\ell_2$-norm and the proposed method on StatPearl corpus with MedCPT. In each plot, we chose three types of targeted information to create poisoned documents. We observed that the proposed method consistently achieves a near-perfect detection rate, while the $\ell_2$-norm methods fail.

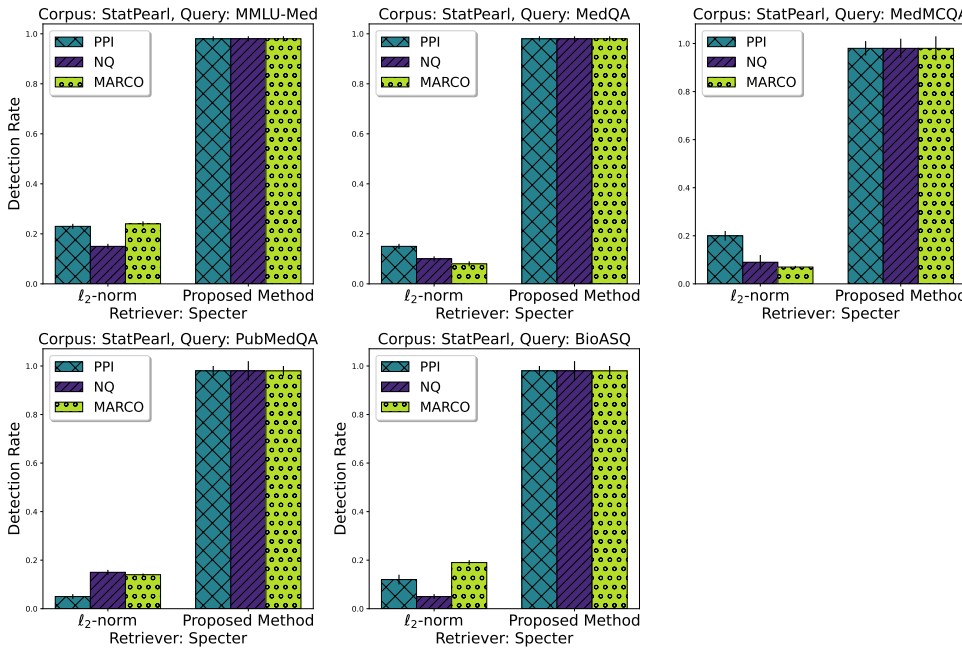

Figure 7: Detection results of the commonly used $\ell_2$-norm and the proposed method on StatPearl corpus with Specter. In each plot, we chose three types of targeted information to create poisoned documents. We observed that the proposed method consistently achieves a near-perfect detection rate, while the $\ell_2$-norm methods fail.

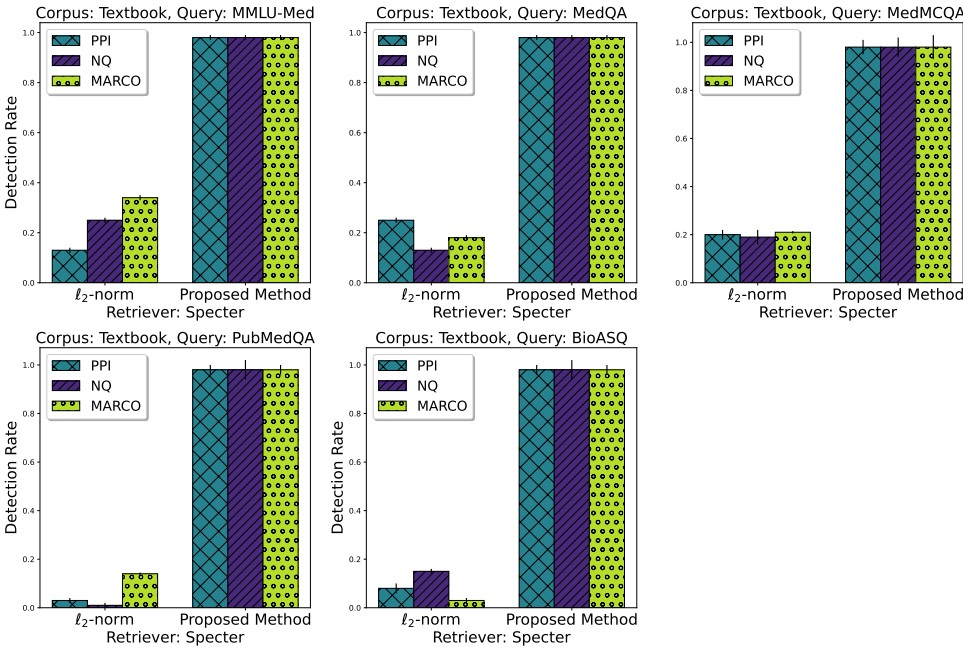

Figure 8: Detection results of the commonly used $\ell_2$-norm and the proposed method on Textbook corpus with Specter. In each plot, we chose three types of targeted information to create poisoned documents. We observed that the proposed method consistently achieves a near-perfect detection rate, while the $\ell_2$-norm methods fail.

## C.2   ON THE SHRINKAGE LEVEL $\beta$

In this section, we provide an ablation study on the choices of $\beta$ used in calculating the proposed distances. We summarize the results in Table 8 below, with standard errors within 0.08. We observed

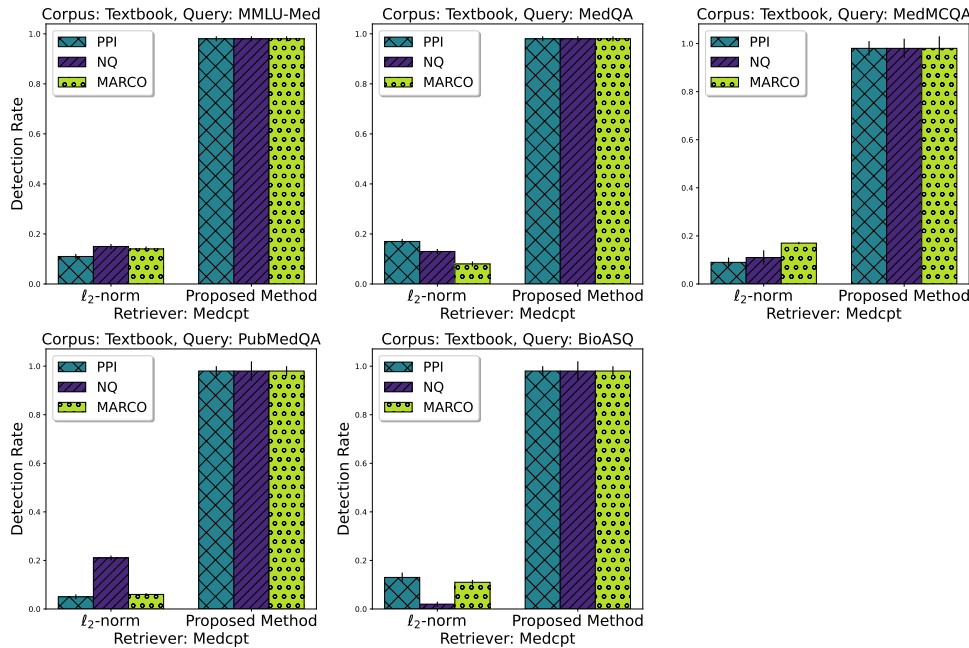

Figure 9: Detection results of the commonly used $\ell_2$-norm and the proposed method on Textbook corpus with MedCPT. In each plot, we chose three types of targeted information to create poisoned documents. We observed that the proposed method consistently achieves a near-perfect detection rate, while the $\ell_2$-norm methods fail.

that as $\beta$ increases, the detection rate begins to decrease. This is reasonable since the covariance tends to shrink towards an identity matrix and hence loses the ability to capture the data's distribution. In practice, we suggest the defender employ cross-validation-based techniques to select the optimal $\beta$ for detection.

Table 7: Top 1 retrieval success rates over 3 corpora, 3 retrievers, 5 query sets, and 5 sets of targeted information. The interpretation of results in each cell adheres the following rule. We use the top-left cell as an example: it represents a success rate of 0.78 achieved using the Corpus: Textbook, Retriever: MedCPT, Query set: MMLU-Med, with PPI as the targeted information.

| Corpus | Retriever | Query | Target Information | | | | |
| | | | PPI | MS-MARCO | NQ | HotpotQA | Diagnostic |
|---|---|---|---|---|---|---|---|
| Textbook | MedCPT | MMLU | 0.74 | 0.80 | 0.81 | 0.78 | 0.81 |
| | | MedQA | 0.98 | 0.98 | 0.98 | 0.98 | 0.99 |
| | | MedMCQA | 0.81 | 0.80 | 0.76 | 0.82 | 0.78 |
| | | PubMedQA | 0.98 | 0.96 | 0.96 | 0.96 | 0.94 |
| | | BioASQ | 0.91 | 0.97 | 0.93 | 0.92 | 0.91 |
| | Specter | MMLU | 0.92 | 0.73 | 0.82 | 0.93 | 0.78 |
| | | MedQA | 0.98 | 0.98 | 0.97 | 0.98 | 0.98 |
| | | MedMCQA | 0.88 | 0.80 | 0.70 | 0.86 | 0.87 |
| | | PubMedQA | 0.92 | 0.82 | 0.92 | 0.91 | 0.91 |
| | | BioASQ | 0.95 | 0.93 | 0.95 | 0.94 | 0.91 |
| | Contriever | MMLU | 0.78 | 0.80 | 0.81 | 0.84 | 0.81 |
| | | MedQA | 1.0 | 1.0 | 1.0 | 1.0 | 0.99 |
| | | MedMCQA | 0.60 | 0.60 | 0.65 | 0.64 | 0.67 |
| | | PubMedQA | 0.78 | 0.80 | 0.82 | 0.80 | 0.80 |
| | | BioASQ | 0.61 | 0.61 | 0.60 | 0.61 | 0.61 |
| StatPearls | MedCPT | MMLU | 0.73 | 0.72 | 0.70 | 0.73 | 0.72 |
| | | MedQA | 0.98 | 0.98 | 0.98 | 0.99 | 0.97 |
| | | MedMCQA | 0.71 | 0.73 | 0.68 | 0.72 | 0.74 |
| | | PubMedQA | 0.93 | 0.92 | 0.91 | 0.90 | 0.91 |
| | | BioASQ | 0.86 | 0.91 | 0.85 | 0.86 | 0.86 |
| | Specter | MMLU | 0.92 | 0.68 | 0.77 | 0.91 | 0.90 |
| | | MedQA | 1.0 | 1.0 | 0.96 | 0.96 | 0.94 |
| | | MedMCQA | 0.84 | 0.62 | 0.70 | 0.79 | 0.78 |
| | | PubMedQA | 0.97 | 0.80 | 0.85 | 0.95 | 0.93 |
| | | BioASQ | 0.91 | 0.80 | 0.87 | 0.93 | 0.92 |
| | Contriever | MMLU | 0.80 | 0.81 | 0.81 | 0.81 | 0.81 |
| | | MedQA | 1.0 | 1.0 | 1.0 | 1.0 | 1.0 |
| | | MedMCQA | 0.61 | 0.65 | 0.63 | 0.66 | 0.61 |
| | | PubMedQA | 0.78 | 0.77 | 0.78 | 0.78 | 0.75 |
| | | BioASQ | 0.58 | 0.56 | 0.58 | 0.58 | 0.61 |
| PubMed | MedCPT | MMLU | 0.72 | 0.74 | 0.71 | 0.73 | 0.71 |
| | | MedQA | 0.99 | 0.99 | 0.98 | 0.99 | 0.99 |
| | | MedMCQA | 0.70 | 0.71 | 0.68 | 0.71 | 0.71 |
| | | PubMedQA | 0.91 | 0.94 | 0.91 | 0.88 | 0.91 |
| | | BioASQ | 0.85 | 0.91 | 0.86 | 0.81 | 0.85 |
| | Specter | MMLU | 0.91 | 0.75 | 0.81 | 0.94 | 0.90 |
| | | MedQA | 0.99 | 0.98 | 0.98 | 0.96 | 0.95 |
| | | MedMCQA | 0.81 | 0.82 | 0.70 | 0.81 | 0.91 |
| | | PubMedQA | 0.76 | 0.82 | 0.85 | 0.91 | 0.96 |
| | | BioASQ | 0.93 | 0.86 | 0.82 | 0.91 | 0.92 |
| | Contriever | MMLU | 0.80 | 0.82 | 0.81 | 0.81 | 0.81 |
| | | MedQA | 1.0 | 1.0 | 1.0 | 1.0 | 1.0 |
| | | MedMCQA | 0.62 | 0.63 | 0.61 | 0.64 | 0.61 |
| | | PubMedQA | 0.76 | 0.77 | 0.74 | 0.76 | 0.71 |
| | | BioASQ | 0.57 | 0.56 | 0.60 | 0.57 | 0.60 |

Table 8: Detection performance on different choices of $\beta$.

| Corpus | Query Set | | | | | |
|---|---|---|---|---|---|---|
| | $\beta = 0.001$ | $\beta = 0.005$ | $\beta = 0.01$ | $\beta = 0.05$ | $\beta = 0.1$ | $\beta = 0.2$ |
| Textbook | .99 | .99 | .99 | .98 | .98 | .97 |
| | .99 | .99 | .99 | .98 | .98 | .97 |
| | .99 | .99 | .99 | .98 | .98 | .98 |
| PubMed | .99 | .99 | .99 | .98 | .96 | .96 |
| | .99 | .99 | .99 | .97 | .96 | .95 |
| | .99 | .99 | .99 | .97 | .97 | .96 |

