# OpenReview forum: "On the Vulnerability of Applying Retrieval-Augmented Generation within Knowledge-Intensive Application Domains"
_ICLR.cc/2025/Conference — ICLR 2025 Conference Withdrawn Submission_

### Official Review · Reviewer_p3F4 · 2024-11-04

**Soundness:** 3
**Presentation:** 3
**Contribution:** 3
**Rating:** 5
**Confidence:** 4

**Summary:**

This paper focuses on the adversarial robustness of RAG, which is a common technique used in LLMs for applications requiring intensive knowledge. The paper proposes a universal poisoning attacks for evaluating the robustness of existing retrieval mechanisms. Based on that, the paper proposes a new defense strategy.

**Strengths:**

1. The paper provides an interesting perspective as universal poisoning attacks for RAG, which sounds new and helps evaluate the vulnerability of existing RAG methods.
2. The analysis of the vulnerability of existing RAG is accompanied by rigid analysis. The paper analyzes the existing pipelines from the angle of orthogonal augmentation property and verifies this property with experiments. Such an analysis makes the claim reasonable.
3. The new defense is reasonable based on the aforementioned analysis. Such a structure makes the paper easy-to-follow.

**Weaknesses:**

The discussion on defense mechanism lacks technical depth. The paper just uses $l_2$-norm based defense as baseline for evaluating the effectiveness of the proposed method. Such a baseline is relatively simple and makes the comparison results not convincing. The design of using distance metrics as a defense is not convincing either based on the analysis in L473-L480. It is not that intuitive for the readers to understand the idea of using this defense yet. The paper's defense strategy is for detection, but it does not include design for post-detection operation, which is more important for a defense view. Additionally, the application field is limited to medical QA in experiments.

**Questions:**

1. Does the proposed defense have any guaranteed performance in detection?
2. What are the cases where the detection is not doing well?

---

### Official Review · Reviewer_8hqi · 2024-11-04

**Soundness:** 2
**Presentation:** 2
**Contribution:** 2
**Rating:** 3
**Confidence:** 4

**Summary:**

This paper investigates the vulnerability of retrieval systems in Retrieval-Augmented Generation (RAG) within knowledge-intensive domains, especially in medical Q&A. It shows that retrieval systems are vulnerable to universal poisoning attacks through extensive experiments across 225 setups. The reasons for this vulnerability are explored, including an orthogonal augmentation property and the low similarity between queries and clean retrieved documents. A new detection-based defense method is developed, achieving excellent detection rates, and the paper also discusses limitations and future research directions.

**Strengths:**

1. Based on an in-depth understanding of the vulnerability of the retrieval system, the article proposes a detection method based on distance metrics (such as Mahalanobis distance) to defend against universal poisoning attacks. This method utilizes the orthogonal relationship between the query and clean documents as well as toxic documents. By regularizing the sample covariance matrix, it can effectively distinguish between toxic documents and clean documents.
2. A large number of experiments have been carried out in the medical Q&A field, covering multiple query sets (such as MMLU - Med, MedQAUS, MedMCQA, PubMedQA, BioASQ - Y/N, etc.), different corpora (Textbook, StatPearls, PubMed), various types of targeted information (including personally identifiable information, medical diagnosis information, etc.), and multiple representative retrievers (Contriever, SPECTER, MedCPT), forming a total of 225 different experimental combinations. Such a comprehensive setup can fully evaluate the performance and vulnerability of the retrieval system under different conditions.
3. The orthogonal augmentation property has been proposed to explain the vulnerability of the retrieval system, providing an angle for understanding the problem and conducting some analyses.

**Weaknesses:**

1. Limitations in application fields: The experiments mainly focus on the medical Q&A field. Although this field is representative, it cannot fully represent all knowledge-intensive application fields. RAG may have different characteristics and security risks in other fields (such as finance, law, etc.), which requires further research and verification.
2. Dependence on retrievers: The effectiveness of the defense method may depend on the characteristics of the retrievers used. Although several common retrievers have been experimented in the paper, for different types of retrievers, the applicability and performance of the method may be affected. For example, the sparse retriever (BM25).
3. Similarity measurement metrics: The inner product is used to calculate the similarity between the query and the retrieved documents in this work. Why not use the cosine similarity to calculate it? The orthogonal augmentation property proposed in the paper actually does not consider the cosine similarity either (because the cosine similarity itself already includes the processing of the angle).
4. Writing problem: When proposing the defense method, the detailed implementation details of the method, the algorithm process, and how it is constructed based on the theoretical foundation are not described in sufficient detail.

**Questions:**

1. On the universality of the orthogonal augmentation property: Does the orthogonal augmentation property mentioned in the paper apply to all types of retrievers and corpora? Are there any special circumstances that would affect the establishment of this property, thereby affecting the effectiveness of the attack and the applicability of the defense method?

2. The extensibility of the defense method: How to extend the proposed defense method to any query set, rather than being limited to specific target query sets? This is crucial for the universality in practical applications.

3. The balance between performance and efficiency: While ensuring the effectiveness of the defense method, how to improve its application efficiency in large-scale data and real-time systems? Is there room for further optimization to meet the performance requirements of practical applications?

4. The choice of similarity metrics: Why not use cosine similarity? Have the results under cosine similarity been tested? (as we already know the drawback of the inner product.)

Grammar and typos:
1.line 50. The author of xxx \citet{}

---

### Official Review · Reviewer_Jnik · 2024-11-04

**Soundness:** 2
**Presentation:** 3
**Contribution:** 2
**Rating:** 5
**Confidence:** 3

**Summary:**

this paper investigates the adversarial robustness of RAG, focusing on the retrieval system. It shows retrieval systems are vulnerable to universal poisoning attacks in medical Q&A. The paper explains this vulnerability and develops a new detection-based defense, which achieves excellent detection rates across various Q&A domains.

**Strengths:**

1. This paper proposes an effective defense method.

2. The experiment is well designed.

**Weaknesses:**

1. The experiments mainly focus on the field of medical Q&A, and the applicability to other application domains has not been fully verified. we suggest testing in legal or financial domains, which also deal with sensitive information and complex queries.

2. There is a lack of more method comparisons. The attack method is single and more relevant method comparisons are needed. Such as PoisonedRAG, or other poisoning attacks from recent literature on RAG security.

3. For large-scale datasets, the distance-based calculation method requires high computing resources. How to balance resources and the accuracy of results? Please provide runtime analyses comparing their method to alternatives, or approaches for improving efficiency, such as dimensionality reduction techniques or approximate nearest neighbor methods.

**Questions:**

1. Although the types of target information used in the experiment cover a variety of situations, for some special medical information with higher professionalism and complexity (such as detailed diagnostic indicators and treatment plans for rare diseases), will the success rate of the attack and the effectiveness of the defense method change? Could you test your method on a specific dataset of complex medical information, if one exists, or propose how you might generate synthetic complex medical data to evaluate this.

2. Why not use more similarity comparisons, such as cosine similarity?

3. In a multilingual environment, will the vulnerabilities of the retrieval system change? Do the semantic expressions and vocabulary usage habits of different languages have different impacts on the universal poisoning attack?

---

### Official Review · Reviewer_9Vn9 · 2024-11-04

**Soundness:** 2
**Presentation:** 2
**Contribution:** 2
**Rating:** 5
**Confidence:** 4

**Summary:**

This paper studies the vulnerabilities of retrieval systems against various poisoning attacks. The authors first conduct analyses on multiple corpus, retrievers, and datasets showing the significant of safety risks in retrieval. Then, they attribute the retriever failure to the previous doc embedding distance metric. Finally, they propose a new metric that can better distinguish between clean and poisoned documents.

**Strengths:**

* This paper studies an important and timely problem, crucial for building responsible and trustworthy retrieval systems.
* The study provides valuable insights into how poisoned document representations behave within the embedding space, such as the orthogonal augmentation property, supported by extensive experimental results.

**Weaknesses:**

* This paper primarily focuses on retrieval systems rather than RAG. The current presentation is somewhat misleading. The title sets expectations for a different scope of study, so a more accurate phrasing would better align with the paper’s actual focus. The authors also make several claims about RAG, suggesting that the proposed attacks could be harmful to RAG systems, yet they do not provide concrete experimental results to substantiate these claims. To strengthen the paper, the authors should either reframe it as an safety study of retrieval systems, similar to Zhong et al., 2023, or include end-to-end experiments on RAG systems as demonstrated in Zou et al., 2024.
* There is a notable lack of discussion and comparison with the highly relevant literature on adversarial attacks and defenses for neural retrievers beyond the related work section. Expanding on this would provide a more comprehensive perspective and strengthen the paper's contribution.
* The experiments focus on an oversimplified setting, where the poisoned documents consist of query variants appended with target adversarial information. This raises two concerns:
  * Will the findings from the analyses hold true for broader attack scenarios?
  * Will the proposed defense be generalizable if the poisoned documents are created using different methods?

**Questions:**

See weaknesses.

---

### Note · Authors · 2024-11-25

**Comment:**

We sincerely thank the reviewers for their time and effort in reviewing our work, as well as the Area Chair for overseeing the process. Given the time constraints, we are unable to complete the additional experiments requested by the reviewer. As a result, we will withdraw the paper. We greatly appreciate their feedback and will incorporate their valuable comments to enhance the manuscript in future revisions.

**Withdrawal Confirmation:**

I have read and agree with the venue's withdrawal policy on behalf of myself and my co-authors.